

# Body height and waist circumference of young Swiss men as assessed by 3D laser-based photonic scans and by manual anthropometric measurements

Claudia Beckmann, Lafi Aldakak, Patrick Eppenberger, Frank Rühli, Kaspar Staub[**] and Nicole Bender[**]

Institute of Evolutionary Medicine, University of Zurich, Zurich, Switzerland
[**] These authors contributed equally to this work.

## ABSTRACT

Overweight and obesity are considered among the major health concerns worldwide. The body mass index is a frequently used measure for overweight and obesity and is associated with common non-communicable diseases such as diabetes type II, cardiovascular diseases and certain cancers. However, the body mass index does not account for the distribution of body fat and relative fat to muscle mass. 3D laser-based photonic full body scans provide detailed information on various body circumferences, surfaces, and volumes as well as body height and weight (using an integrated scale). In the literature, body scans showed good feasibility, reliability, and validity, while also demonstrating a good correlation with health parameters linked to the metabolic syndrome. However, systematic differences between body scan derived measurements and manual measurements remain an issue. This study aimed to assess these systematic differences for body height, waist circumference, and body mass index using cross-sectional data from a homogenous sample of 52 young Swiss male volunteers. In addition to 3D laser-based photonic full body scans and correlative manual measurements, body fat distribution was assessed through bioelectrical impedance analysis. Overall, an excellent correlation was found between measurements of waist circumference and body mass index, and good correlation between body mass index and total fat mass, as well as between waist circumference and visceral fat mass as assessed by bioelectrical impedance analysis. Volunteers were shorter in height measured by body scan when compared to manual measurements. This systematic difference became smaller when volunteers stood in the scanner in a completely upright position with their feet together. Waist circumference was slightly smaller for manual measurements than for body scan derived values. This systematic difference was larger in overweight volunteers compared to leaner volunteers.

## INTRODUCTION

Overweight and obesity are major global health concerns, and worldwide more than 1.9 billion adults were overweight in 2016 (*WHO, 2017*). The body mass index (BMI, weight in

Corresponding author
Nicole Bender,
nicole.bender@iem.uzh.ch

kilograms divided by squared height in meters) is a frequently used measure for overweight and obesity and is associated with common non-communicable diseases such as diabetes type II, cardiovascular diseases and certain cancers (*WHO, 2018*). However, the BMI does not account for the distribution of body fat and relative fat to muscle mass. Waist circumference (WC) and waist-to-hip ratio are considered to be better than the BMI to predict CVD (*Kjaer et al., 2015*; *WHO, 2008*). However, it is still unclear which measures best correlate with disease risk (*Ashwell, Gunn & Gibson, 2012*; *Kahn & Bullard, 2016*; *Lam et al., 2015*). Other techniques used at a population level like Bioimpedance Analysis (BIA) offer and quick and safe measurement of height, weight, BMI, total fat mass, visceral fat mass, and muscle mass, but measures like WC have to be taken classically with a tape and values have to be entered manually in the BIA machine. Furthermore, BIA seems to be less precise in measuring visceral fat mass than other techniques such as the reference standards, computed tomography (CT) and magnetic resonance imaging (MRI) (*Murphy et al., 2019*). Other circumferences and ratios derived from them are not assessed. Such measurements have therefore their own limitations and new approaches to quantify obesity and categorize the corresponding health risks are therefore necessary.

3D laser-based photonic full body scans create a detailed surface image of the human body consisting of up to 300 data points per $cm^2$ within 10–12 s. The BS technique provides detailed information on whole body or body-part circumferences, surfaces and volumes as well as body height and weight (using an integrated scale) in a way that is not only fast and non-invasive but also comfortable for scanned individuals. Scan data are increasingly used for the acquisition of clinically relevant anthropometric measurements by comparing 3D scan data with data from classical anthropometry (*Bretschneider et al., 2009*; *Koepke et al., 2017*; *Lin et al., 2004*; *Olivares et al., 2007*; *Wells et al., 2015*). Recent studies comparing the body scan technique with classical anthropometry have demonstrated the applicability of the scan technique in an epidemiological context (*Jaeschke, Steinbrecher & Pischon, 2015*; *Kuehnapfel et al., 2016*). In these studies, body scans showed good feasibility, reliability, and validity, and correlations with health parameters linked to the metabolic syndrome were comparable to studies using manual measurement techniques (*Jaeschke, Steinbrecher & Pischon, 2015*).

One issue with body scan studies are systematic differences between body scan derived measurements and manual measurements, which can be considered as the standard. Height derived from scans is slightly less than manually measured height, and different circumferences are either systematically larger or smaller, when derived from scans, as opposed to manual measurement techniques, in various studies (*Domina, Heuberger & MacGillivray, 2008*; *Jaeschke, Steinbrecher & Pischon, 2015*; *Janssen et al., 2002*; *Koepke et al., 2017*; *Wang et al., 2006*). Possible factors influencing such differences are variability of posture, the degree of inspiration, and movement of volunteers while standing in the scanner. Also, there are technical differences between different scanner systems in terms of hardware precision, automatic landmark setting, and measurement output of the scanner software (*Schwarz-Müller, Marshall & Summerskill, 2018*). Regarding different body circumferences such as waist or hip circumference, systematic differences between scans and manual measurements can be explained by a slightly tighter fitting or slightly

different positioning of the measurement tape, compared to scan derived circumferences (*Koepke et al., 2017*; *Wang et al., 2006*).

This study aimed to further assess systematic differences between body scans and manual measurements in height, WC, and BMI, with a goal to further validate scan derived measurements in future population based studies. We also aimed to assess the correlation between BMI, derived from different scanner or manual measurements, and total body fat mass, derived from BIA. Similarly, we aim to analyse the correlation between WC, derived from different scanner and manual measurements, and visceral fat mass, derived from BIA.

## METHODS

As part of a larger research project, a cross-sectional study assessing waist circumference, body height and weight, as well as body composition was conducted on 52 Swiss Armed Forces recruits during their basic training. As this is a purely methodological, and not an epidemiological study, we consider a (homogenous) sample size of 52 to be sufficient in order to show systematic differences between measurement techniques. The study took place in Neunkirch in the Canton of Schaffhausen, Switzerland, in August 2017. Study volunteers were all male Swiss Armed Forces recruits aged 19–23 (19 years old: 34.6%, 20 years old: 38.5%, 21 years old: 19.2%, 22 years old: 7.7%, mean age: 20.5 years). Volunteers came from different parts of Switzerland and were not selected for socioeconomic status or other demographic variables. This study was conducted with institutional review board approval (BASEC No. 2016-01625), and participation was voluntary. All volunteers were briefed in written form at the beginning of the study and again orally before the examination. All volunteers signed a detailed informed consent form.

Body scanner derived measurements were acquired using a semi-mobile Body Scanner (VITUSbodyscan,Human Solutions, Kaiserslautern, Germany). This scanner model is equipped with four eye-safe lasers, eight cameras, and acquires up to 300 data points per $cm^2$ as a 3D point cloud, based on optical triangulation. This type of scanner showed to be reliable and precise in its measurements (*Koepke et al., 2017*). The scanner operating software (Anthroscan, Human Solutions, Kaiserslautern, Germany) calculates more than 150 automatic standard measurements, including height, and a large number of distances and circumferences. The scanner was calibrated according to the manufacturer's instructions at the beginning of the data collection days when body scans and manual measurements were acquired. Volunteers were measured in two positions, once standing straight in an upright position with their feet together (same position as during the manual height measurement with the stadiometer) and another in a standardized position specified by the scanner manufacturer (standing in an upright position, both feet positioned on marks on the scanner platform (spaced approximately 30 cm apart), arms slightly bent at the elbow and held slightly away from the body, head in accordance to the Frankfurt Horizontal Plane). All volunteers were briefed before each scan regarding exact positioning on the platform. They were asked to hold their breath after exhalation for the scans, which

was about 10 s per scan. Volunteers wore only form-fitting underpants and a tight-fitting bathing cap. See Figs. S1a and S1b for the two scan positions.

For standard WC measurements, a hand-held tape of stretch-resistant quality and automatic retraction was used (seca 201, Seca AG, Reinach, Switzerland). All WC measurements were performed by one of the authors trained in WC measurements (NB) (*Koepke et al., 2016*; *Staub et al., 2018*). The tape measurement position was chosen according to WHO guidelines, at midlevel between the lowest palpable point of the rib cage and the highest palpable point of the iliac crest (*WHO, 2008*). The measurement level was marked with a pen in order to make it visible in subsequently performed body scan acquisitions. Due to a tight schedule set by the Swiss Armed Forces, manual WC measurements were only carried out once for each volunteer. Height measurements were carried out with a standard stadiometer (seca 274, Seca AG, Reinach, Switzerland). Volunteers stood straight in an upright position with their feet together, their back and feet against the stadiometer, and head positioned in accordance with the Frankfurt Horizontal Plane.

To assess body composition, bioelectric impedance analysis (BIA) was utilized, as time constraints and the need for non-invasive methods did not allow for other assessment methods such as dual-energy X-ray absorptiometry (DXA) or magnetic resonance imaging (MRI). We used a medical 8-point body composition analyzer (Seca mBCA 515, Seca AG, Reinach, Switzerland) measuring weight and calculating the amount of whole-body fat mass and visceral fat mass, muscle mass, and intracellular and extracellular water in the body based on bioelectric impedance measurements across a total of four pairs of electrodes placed on both hands and both feet. Whole body and body part composition was calculated based on mathematical algorithms using the integrated software. This software was validated in different settings and with different ethnic groups (*Bosy-Westphal et al., 2017*; *Bosy-Westphal et al., 2013*; *Day et al., 2018*). For BIA, study volunteers stood barefoot on the two pairs of foot-electrodes and placed each hand on one of the two pairs of hand-electrodes.

A dataset using measurements from all described sources was compiled. Height and WC from manual measurements were included. From body scans, height in a straight position and standard position were included. Furthermore, an automatically calculated WC from the scans and a second WC measurement from the scans that was manually adjusted in the software to the WHO measurement point were included. To obtain this second WC scanner measurement, the measurement line of the software was manually adjusted to match the pen mark of the manual WHO measurement level on the scans (see Fig. S1c). Additionally, weight (from an incorporated electronic scale, Seca AG, Reinach, Switzerland), relative whole-body fat mass (%), and visceral fat mass (l) from BIA measurements were included. There was one missing body scan in straight position from one study participant. Therefore, for some statistics, $N = 51$ was used instead of $N = 52$.

Volunteers' BMI was calculated from their weight divided by the square of their height in meters in accordance to WHO guidelines (*WHO, 2018*). Volunteers were classified in different BMI subgroups, volunteers with BMI <18.5 kg/m2 as underweight, volunteers with BMI 18.5–24.9 kg/m2 as normal weight, volunteers with BMI ≥ 25.0–29.9 kg/m2 as

**Table 1 Measurements.**

| Used terms | Explanation |
| --- | --- |
| Manual WC | WC measured with classic manual anthropometry (tape) |
| Automatic scan WC | WC measured with the scanner at automatic scanner position |
| Adjusted scan WC | WC measured with the scanner adjusted at WHO point |
| Manual height | Height measured with classic manual anthropometry (stadiometer) |
| Standard scan height | Height measured with the scanner in standard body posture |
| Straight scan height | Height measured with the scanner in straight body posture |
| Manual BMI | BMI calculated with height from classic manual anthropometry |
| Standard scan BMI | BMI calculated with height from the scanner in standard body posture |
| Straight scan BMI | BMI calculated with height from the scanner in straight body posture |

overweight, and volunteers with BMI ≥ 30.0 kg/m2 as obese (*WHO, 2017*). In accordance with WHO guidelines, volunteers were categorized for their risk of metabolic complications (RMC) according to their WC. WC <94.0 cm is associated to a low RMC, WC 94.0–102.0 cm is associated to an increased RMC, and WC >102.0 cm is associated to a substantially increased RMC (*WHO, 2008*).

We included the following WC measurements in the study: the manual WC measurement, the automatic scanner measurement in the standard position, and the adjusted scanner measurement at the WHO measurement point. For height, we included the manually measured height, height measured with the scanner in a straight position ($N = 51$), and height measured with scanner in standard position. We calculated BMI using weight from the internal scale of the BIA and height from the manual measurement as well as using height from the scan in the straight position and the standard position. See Table 1 for an overview of all measurements considered and abbreviations used.

## Statistical analysis

Descriptive statistics were calculated for all above-listed measures of WC, height, and BMI. The distributions for WC and BMI were not entirely symmetrical (Fig. S2), and logarithmic transformation did not considerably change their shape. Therefore, non-parametric and parametric methods were applied to analyse the data. Differences between different WC, height, and BMI measures were tested using Wilcoxon signed-rank tests and paired $t$-tests. To assess the agreement between the different measurements for WC, height, and BMI, and to compare both measurement methods (scanner vs. manual), Lin's correlations coefficients (CCC) (*Lin, 1989*) were used, Pearson's as well as Spearman rank correlations. Kappa coefficients (*Altman, 1999*) were used to assess the classification agreement according to official WHO BMI categories for overweight/obesity as well as increased health risks for WC. Agreements, correlations, and intra- and inter-methods comparisons were visualized using scatterplots and Bland Altman Plots (*Bland & Altman, 1999*). For the Bland Altman

plots the batplot package in Stata (Version 14.1) was used, which additionally tests the difference between the methods for trend. To compare effect sizes between methods a linear regression was performed to assess the association between BMI and relative whole-body fat mass, divided into BMI subcategories, separately for the different BMI measures. Similarly, a linear regression was performed to assess the association between WC and visceral fat mass, divided into WC subcategories, separately for the different WC measures.

## RESULTS

Visual evaluation of WC measure scatterplots showed an increasing deviation with larger WCs when manually measured WC was compared to automatic scan WC and to manually adjusted scan WC (Figs. 1A and 1C). This increasing deviation is also represented as an increasing difference between the measures with an increasing average of the measures (Figs. 1B and 1D). The largest difference ($-1.64$ cm, paired $t$-test $p < 0.01$) between WC measurements was observed between manual WC (mean 81.65 cm, SD = 9.04) and adjusted scan WC (mean 83.29 cm, SD = 9.95) (Table 2). The smallest and not significant difference ($-0.13$ cm, paired $t$-test $p = 0.376$) was observed between automatic scan WC (mean 83.15 cm, SD = 9.73) and adjusted scan WC (mean 83.29 cm, SD = 9.95). Agreement between automatic scan WC and adjusted scan WC was good (kappa=0.68, agreement = 92.88%) and very good between manual WC and automatic scan WC ($k = 0.91$, agreement = 98.08%). Correlation was very high for all WC measurements (CCC >0.96). For all results see Tables 2–5. The results from the non-parametric tests were very similar (Table 3 and Table 5).

For height, visually there was a constant small difference between manually measured height and standard position scan height as well as between standard position scan height and vertical position scan height. This difference was minimal between manual height and vertical scan height (Figs. 2A, 2C, and 2E). The smallest height measurements difference (+0.20 cm, paired $t$-test $p = 0.009$) was between manual height (mean 178.69 cm, SD = 6.82) and straight scan height (mean 178.18 cm, SD = 6.55) (Table 2). The largest difference (+0.77 cm, paired $t$-test $p < 0.001$) was between manual height (mean 178.69 cm, SD = 6.82) and standard scan height (mean 177.92 cm, SD = 6.78) (Table 2). There were significant differences between all compared height measurements (Table 2). The correlation was very high for the manually measured height, and both scanned height measurements (CCC > 0.98). The results from the non-parametric tests were again very similar (Tables 2 and 3).

For BMI, the inspection of Fig. 3 revealed a strong association between BMI calculated with manual height and BMI calculated with scan height in a straight posture. The visual association was slightly deviating for manual BMI versus standard scan BMI, and for standard scan BMI versus straight scan BMI, with increasing BMI (Figs. 3A, 3C and 3E). All calculated BMI differences were highly significant (Table 2). The smallest difference ($-0.06$, paired $t$-test $p = 0.007$) was between manual BMI (mean 23.99, SD = 4.01) and straight scan BMI (mean 24.09, SD = 4.06). The largest difference ($-0.22$, paired $t$-test $p < 0.001$) was between manual BMI (mean 23.99, SD = 4.01) and standard scan BMI

## Waist Circumference

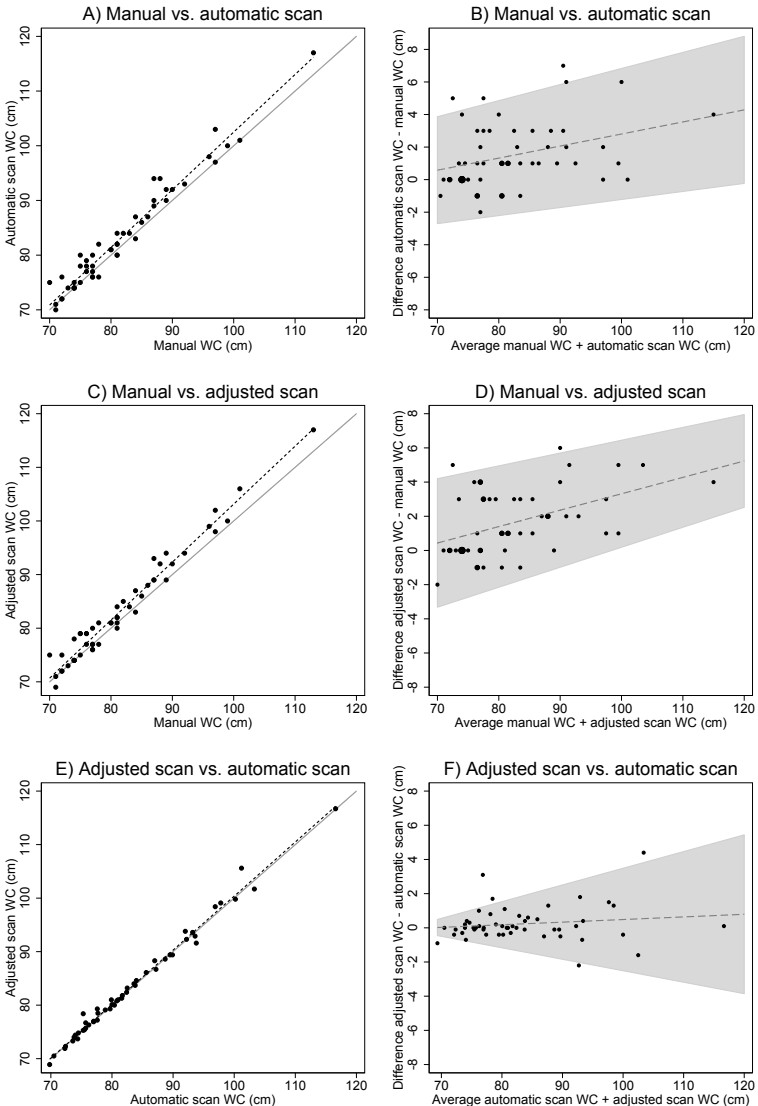

**Figure 1** **Associations between different waist circumference measures.** (A), (C) and (E) show scatter plots of different WC measures (solid line: x=y, dotted line: linear regression). (B), (D) and( F) show Bland Altman plots (the solid line including 95% confidence interval originates from the test for trend using linear regression). There is an increasing deviation between manual measurements and scanner measurements with increasing WC.

(mean 24.20, SD = 4.09). The correlation was excellent for all BMI measures (Kappa = 1.0, agreement 100%, CCC >0.99). For all results see Tables 2–5.

There was a strong correlation between the different BMI measures and relative fat mass (Spearman Rho ≥ 0.86, Table 6, Fig. 4A). The WHO subcategories of BMI correlated similarly with relative fat mass in all three BMI measures (Fig. 5A). Compared to normal weight volunteers, underweight volunteers measured with classic anthropometry showed

**Table 2  Descriptive statistics for WC, height, and BMI.** (A) N, mean, standard deviation (SD), and median. (B) Wilcoxon signed rank test, paired $t$-test.

**(A)**

|  | N | Mean | SD | Median |
|---|---|---|---|---|
| **WC** | | | | |
| Automatic scan WC | 52 | 83.15 | 9.73 | 80.50 |
| Adjusted scan WC | 52 | 83.29 | 9.95 | 81.00 |
| Manual WC | 52 | 81.65 | 9.04 | 80.00 |
| **Height** | | | | |
| Standard scan height | 52 | 177.92 | 6.78 | 176.10 |
| Straight scan height | 51 | 178.18 | 6.55 | 176.40 |
| Manual height | 52 | 178.69 | 6.82 | 177.25 |
| **BMI** | | | | |
| Standard scan BMI | 52 | 24.20 | 4.09 | 23.81 |
| Straight scan BMI | 51 | 24.09 | 4.06 | 23.54 |
| Manual BMI | 52 | 23.99 | 4.01 | 23.55 |

**(B)**

|  | Wilcoxon signed rank test | | Paired $t$-test | |
|---|---|---|---|---|
|  | z | p | Delta (cm) | p |
| **WC** | | | | |
| Manual WC vs. scan WC | −4.54 | <0.001 | −1.50 | <0.001 |
| Manual WC vs. adjusted scan WC | −4.71 | <0.001 | −1.64 | <0.001 |
| Scan WC vs. adjusted scan WC | −1.06 | 0.295 | −0.13 | 0.376 |
| **Height** | | | | |
| Manual height vs. standard scan height | 5.56 | <0.001 | 0.77 | <0.001 |
| Manual height vs. straight scan height | 2.98 | 0.003 | 0.20 | 0.009 |
| Standard scan height vs. straight scan height | −5.08 | <0.001 | −0.55 | <0.001 |
| **BMI** | | | | |
| Manual BMI vs. standard scan BMI | −5.56 | <0.001 | −0.22 | <0.001 |
| Manual BMI vs. straight scan BMI | −3.02 | 0.003 | −0.06 | 0.007 |
| Standard scan BMI vs. straight scan BMI | 5.12 | <0.001 | 0.16 | <0.001 |

−6.6% (95% CI [−13.8–0.5]) less body fat mass. Overweight volunteers showed 11.3% (95% CI [8.2–14.3]) more body fat than normal weight volunteers, and obese volunteers showed 20.5% (95% CI [14.5–26.4]) more body fat than normal weight volunteers (Fig. 5A). The results for BMI calculated with scan height in the standard position and in the straight position were very similar. See Fig. 5 and Table S1 for all results.

There was a strong correlation between the different WC measures and visceral adipose tissue (Spearman Rho ≥ 0.79, Table 5), but manual WC deviated increasingly from automatic scan WC and adjusted scan WC with increasing WC (Fig. 4B). For WC subcategories vs. visceral adipose tissue, the correlation was more different for manual WC than for the other WC measures, when WC was more than 102.0 cm (Fig. 5B). Compared to volunteers with WC <94.0 cm (low RMC) volunteers with WC 94.0–102.0 cm (increased RMC) measured with classic anthropometry had 1.94 litre (95% CI [1.53–2.34]) more

**Table 3** Correlations between different measurements of WC, height, and BMI.

|  | CCC | R | C_b | Rho |
|---|---|---|---|---|
| **WC** |  |  |  |  |
| Manual WC vs. scan WC | 0.964 | 0.979 | 0.985 | 0.964 |
| Manual WC vs. adjusted scan WC | 0.964 | 0.983 | 0.981 | 0.967 |
| Scan WC vs. adjusted scan WC | 0.994 | 0.995 | 1.000 | 0.994 |
| **Height** |  |  |  |  |
| Manual height vs. standard scan height | 0.988 | 0.995 | 0.993 | 0.989 |
| Manual height vs. straight scan height | 0.996 | 0.997 | 0.999 | 0.992 |
| Standard scan height vs. straight scan height | 0.992 | 0.996 | 0.996 | 0.992 |
| **BMI** |  |  |  |  |
| Manual BMI vs. standard scan BMI | 0.997 | 0.999 | 0.998 | 0.998 |
| Manual BMI vs. straight scan BMI | 0.999 | 0.999 | 1.000 | 0.998 |
| Standard scan BMI vs. straight scan BMI | 0.998 | 0.999 | 0.999 | 0.998 |

Notes.

CCC, Lin's concordance correlation coefficient; R, Pearson's correlation coefficient; C_b, bias correction factor; Rho, Spearman correlation coefficient.

**Table 4** WHO categories of manual BMI, standard scan BMI, and straight scan BMI. WHO categories for manual WC, automatic scan WC, and adjusted scan WC.

| Manual BMI (kg/m2) | N | Standard scan BMI (kg/m2) | N | Straight scan BMI (kg/m2) | N |
|---|---|---|---|---|---|
| <18,5 | 2 | <18,5 | 2 | <18,5 | 2 |
| 18,5 - 24,9 | 32 | 18,5–24,9 | 32 | 18,5–24,9 | 32 |
| 25,0 - 29,9 | 15 | 25,0 - 29,9 | 15 | 25,0 - 29,9 | 15 |
| ≥30,0 | 3 | ≥30,0 | 3 | ≥30,0 | 3 |

| Manual WC (cm) |  | Automatic scan WC (cm) |  | Adjusted scan WC (cm) |  |
|---|---|---|---|---|---|
| <94.0 | 46 | <94.0 | 46 | <94.0 | 44 |
| 94.0–101,9 | 5 | 94.0–101,9 | 4 | 94.0–101,9 | 6 |
| ≥102.0 | 1 | ≥102.0 | 2 | ≥102.0 | 2 |

**Table 5** Kappa coefficients and % agreement between categories of different measures of WC and BMI.

|  | Kappa | % Agreement |
|---|---|---|
| **WC** |  |  |
| Manual WC vs. automatic scan WC | 0.9081 | 98.08% |
| Manual WC vs. adjusted scan WC | 0.7593 | 94.23% |
| Automatic scan WC vs. adjusted scan WC | 0.681 | 92.31% |
| **BMI** |  |  |
| Manual BMI vs. standard scan BMI | 1.000 | 100% |
| Manual BMI vs. straight scan BMI | 1.000 | 100% |
| Standard scan BMI vs. straight scan BMI | 1.000 | 100% |

# Height

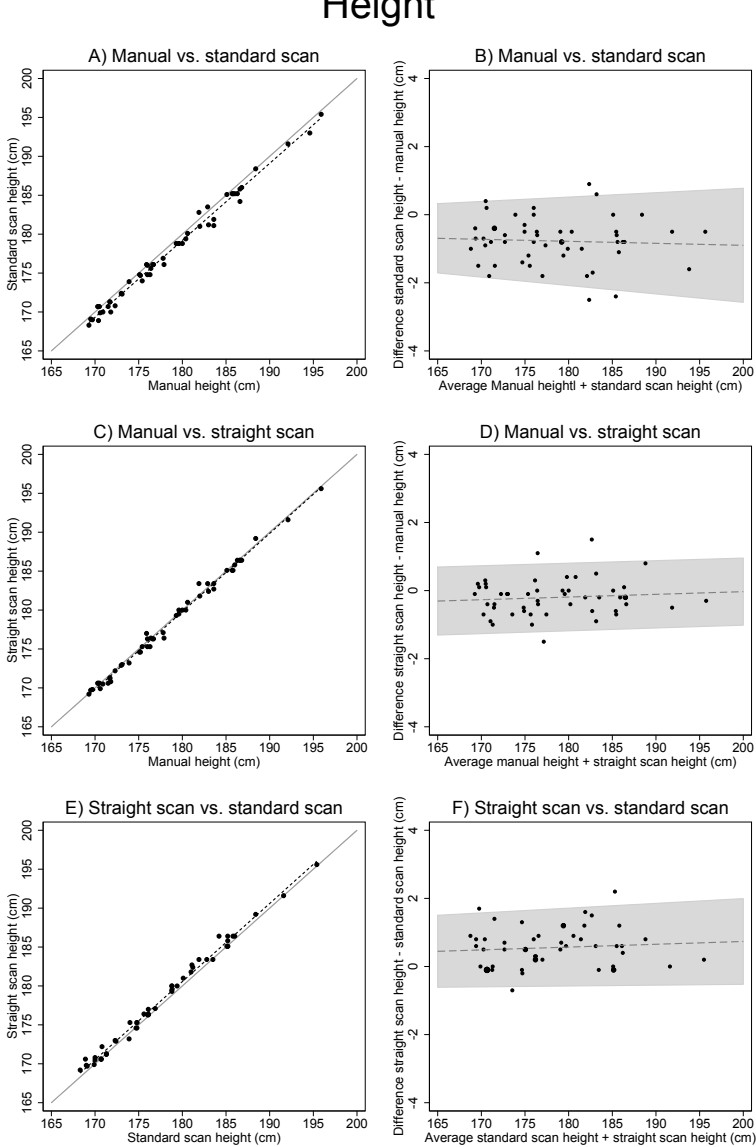

**Figure 2  Associations between different height measurements.** (A), (C) and (E) show scatter plots of different height measures (solid line: x=y, dotted line: linear regression). (B), (D) and (F) show Bland–Altman Plots (the solid line including 95% confidence interval originates from the test for trend using linear regression). (A) shows the systematic difference between manual height measurement and scanner measurement in the standard scanner position.

visceral fat mass. Volunteers with a WC >102 cm (substantially increased RMC) had 4.57 litre (95% CI [3.7–5.44]) more visceral fat mass than volunteers with low risk (WC < 94.0 cm).

## BMI

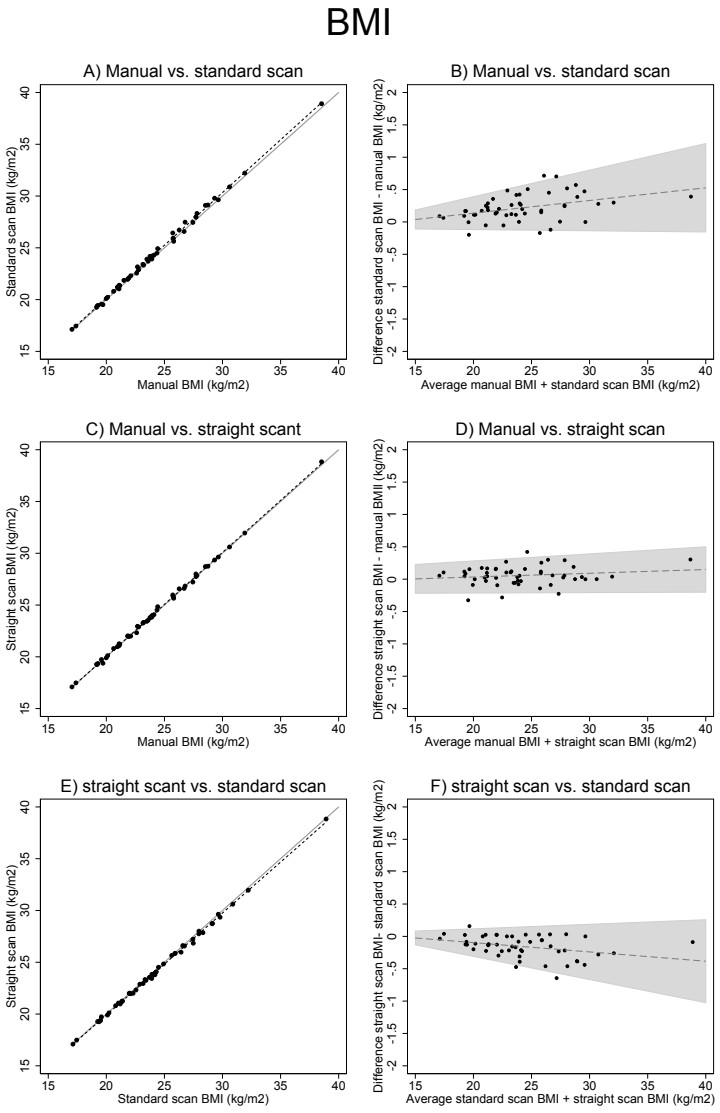

**Figure 3  Associations between different BMI measurements.** (A), (C) and (E) show scatter plots of different BMI measures (solid line: x=y, dotted line: linear regression). (B), (D) and (F) show Bland Altman plots (the solid line including 95% confidence interval originates from the test for trend using linear regression).

# DISCUSSION

The presented study evaluated whether the positioning of volunteers in the scanner influenced previously observed differences between height measurements derived from body scans and manually measured height. We confirmed that height was less when measured by the scanner, compared to classical anthropometry. When volunteers stood in the scanner in a completely upright position with their feet together, this difference decreased smaller. Nevertheless, for height measurements, a small difference remained between the scanner and manual classical anthropometry for the same body position.

**Table 6** **Pearsons (R) and Spearman (Rho) correlation coefficients between measures of (A) BMI and relative fat mass, and between measures of (B) WC and visceral fat mass.**

**(A) Relative fat mass**

| BMI | R | Rho |
| --- | --- | --- |
| Manual BMI | 0.884 | 0.863 |
| Straight scan BMI | 0.883 | 0.861 |
| Standard scan BMI | 0.881 | 0.860 |

**B) Visceral adipose tissue**

| WC | R | Rho |
| --- | --- | --- |
| Manual WC | 0.919 | 0.824 |
| Automatic scan WC | 0.897 | 0.790 |
| Adjusted scan WC | 0.904 | 0.808 |

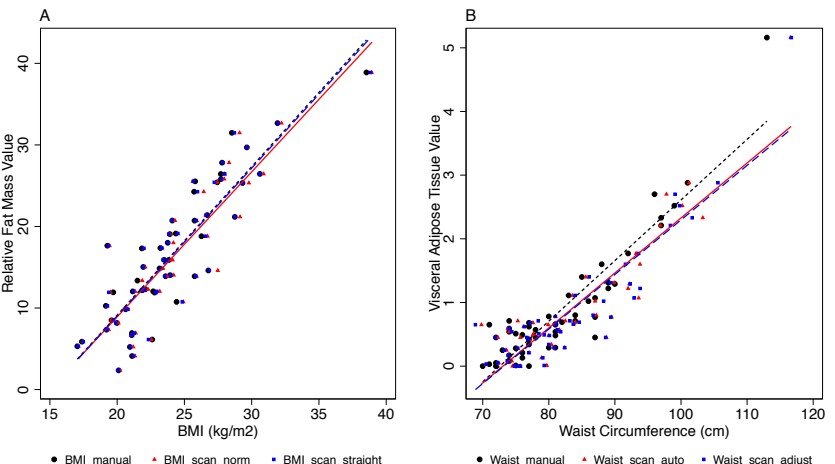

**Figure 4** **Associations between BMI and relative fat mass, and between WC and visceral fat mass.** A, Associations between BMI and relative fat mass (%). The different BMI measures are given in different colours and symbols, and linear regression lines. B, Associations between WC and visceral fat mass (l). Different WC measures are given in different colours and symbols, and linear regression lines. The manual measurements are increasingly deviating from the scanner measurements with increasing WC.

Interestingly, other studies found that height was greater in the scanner than measured with classical anthropometry (*Jaeschke, Steinbrecher & Pischon, 2015*; *Kuehnapfel et al., 2016*). Several factors could explain our findings. In classical anthropometry, the stadiometer behind the volunteers may encourage the volunteers to stand straighter than without the stadiometer in the scanner (*Koepke et al., 2017*). Additionally, posture variation in the scanner could also have influenced height; in fact, other studies showed relevant differences due to posture variation (*Schwarz-Müller, Marshall & Summerskill, 2018*; *Tomkinson & Shaw, 2013*; *Wells et al., 2015*).

As expected and similarly to other studies, we found a strong correlation for WC measured by classical manual anthropometry and scanner derived measurements

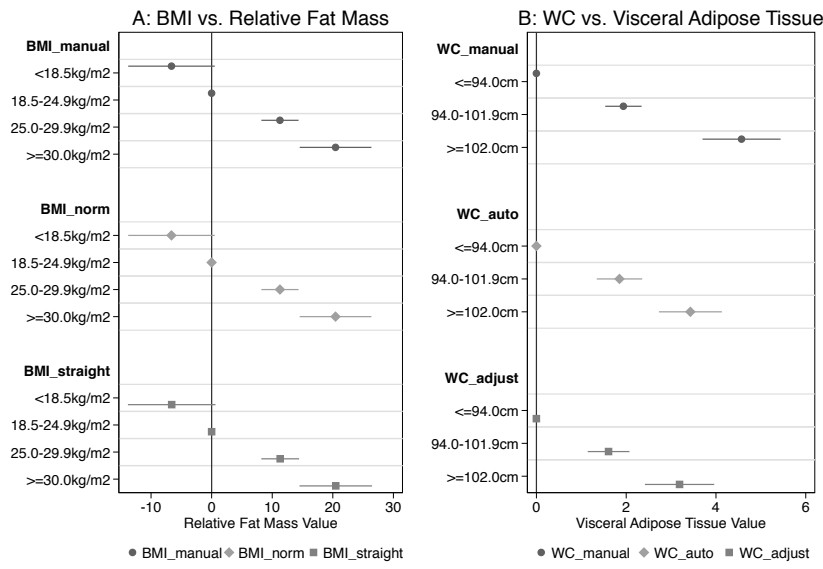

**Figure 5** **Coefficients from regression analysis for categories of BMI vs relative fat mass and for categories of WC vs visceral fat mass.** A, Coefficients and 95% confidence intervals from regression analyses of BMI categories and relative fat mass (%), for the three different measurements. B, Coefficients and 95% confidence intervals from regression analyses of WC categories and visceral fat mass (l), for the three different measurements. The manual WC measuements are increasingly deviating from the scanner measurements with increasing WC.

(*Koepke et al., 2017*). However, WC was systematically slightly smaller if measured with manual anthropometry than derived from the scanner. A possible explanation for this difference is that volunteers tend to pull their stomach in for manual measurements (*Jaeschke, Steinbrecher & Pischon, 2015*). Other studies showed that arm posture had a significant influence on WC measurements (*Lennie et al., 2013*; *Lu, Wang & Mollard, 2010*). In our study arm posture was different in the scanner (slightly bent elbows and slightly separated from the body, but not too far away in order not to leave the measuring area) as in manual WC, where volunteers had to hold their arms further away from the trunk due to the measurement procedure. The difference between manual WC and scanner derived measurements was not explicable by a different measuring level, as manual WC deviated more from adjusted scan WC than expected, while there was no significant difference between automatic scan WC and adjusted scan WC. This means that at the population level automatic scan WC is reliable concerning the measuring level and can be equalized to the WHO measuring level.

Interestingly, manual WC deviated more from scanner measures in obese volunteers compared to lean volunteers. Other studies showed low reliability of manual WC measurements in obese subjects (*Kuehnapfel et al., 2016*; *Verweij et al., 2013*). It was more challenging to control tape position when WC was larger and more difficult to identify the WHO measurement level in obese volunteers correctly.

The scanner derived measures correlated similarly to total fat and visceral fat as the manually derived measures, probably being even superior when it comes to the correlation

between waist circumference and visceral fat in heavily obese people. As just discussed for waist circumference, this might be due to the difficulty in measuring manually the waist circumference in very obese people. Overall the scanner offers a reliable tool to assess a number of anthropometric measures in a fast and safe way. Furthermore, as Table 4 showed, the difference in categorizations of BMI and WC between scanner and manual measurements was absent or minimal. The scanner can therefore be regarded as an alternative tool to categorize patients into BMI and WC categories, if scans are taken for any reason.

Several techniques are used to reconstruct body segment volumes, body segment parameter differences, or segment inertial properties, such as 3D body scanners (*Norton, Donaldson & Dekker, 2002*), infrared scanners (*Smith & Bull, 2018*), magnetic resonance imaging (*Cheng et al., 2000*) or dual energy X-ray absorptiometry (DEXA) (*Durkin & Dowling, 2003*). The different techniques have their advantages and disadvantages, depending on the specific research target. For epidemiological research on anthropometrical measures, a 3D full body scanner offers a rapid, safe and reliable alternative.

This study is the first to compare different postures and different measuring levels to assess height and WC with a 3D body scanner. An advantage for the internal validity of this study was the homogenous study group, namely same age, and sex, which reduced physical inter-individual variability apart from BMI. Limitations for the external validity of this study were the small sample size, and the specificity of the sample, particularly regarding sex and age. Due to time constraints, all measurements were only carried out once per volunteer. One trained researcher did all manual WC measurements. Therefore, we could exclude inter-observer bias, but an intra-observer bias cannot be excluded (*Koepke et al., 2017*).

## CONCLUSION

The body scanner measurement technique seems to be reliable for height and WC and the compared measurement techniques correlated well. However, the small difference in height measurements that also persists after postural correction must be further investigated. Likewise, the observed deviation between manually measured WC and body scan derived values, which systematically increased with increasing BMI of volunteers, must be further investigated. Additionally, extensive studies with volunteers of both sexes, various ages, and different BMIs are required to further evaluate the differences between scanner derived measurements and classic anthropometrics in order to assess the applicability of the scanner technique at a population level.

## ACKNOWLEDGEMENTS

The authors are especially thankful to Andreas Stettbacher (Chief Medical Surgeon), Franz Frey, Alexander Faas, Martino Ghilardi, Marco Müller, and Yvanka Jerkovic from the Swiss Armed Forces for their tremendous (logistic) support. We also thank the IEM collaborators Nikola Koepke, Lena Öhrström, Gulfirde Akgül, Anne Lehner and Nakita Frater for helping to collect the data, and Abagail Breidenstein for English editing.

### Funding

This work was supported by the Mäxi Foundation, Zurich, Switzerland. The funders had no role in study design, data collection and analysis, decision to publish, or preparation of the manuscript.

### Grant Disclosures

The following grant information was disclosed by the authors:
Mäxi Foundation, Zurich, Switzerland.

### Competing Interests

The authors declare there are no competing interests.

### Author Contributions

- Claudia Beckmann performed the experiments, analyzed the data, prepared figures and/or tables, authored or reviewed drafts of the paper, approved the final draft.
- Lafi Aldakak and Patrick Eppenberger performed the experiments, authored or reviewed drafts of the paper, approved the final draft.
- Frank Rühli conceived and designed the experiments, authored or reviewed drafts of the paper, approved the final draft, funding, supervision.
- Kaspar Staub conceived and designed the experiments, performed the experiments, analyzed the data, contributed reagents/materials/analysis tools, prepared figures and/or tables, authored or reviewed drafts of the paper, approved the final draft.
- Nicole Bender conceived and designed the experiments, performed the experiments, authored or reviewed drafts of the paper, approved the final draft.

### Human Ethics

The following information was supplied relating to ethical approvals (i.e., approving body and any reference numbers):

This study was conducted with institutional review board approval of the canton of Zurich, Switzerland (BASEC No. 2016-01625).

### Data Availability

The raw data are available in Table S2.

### Supplemental Information

Supplemental information for this article can be found online at http://dx.doi.org/10.7717/peerj.8095#supplemental-information.

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
