# Peer review of "Body height and waist circumference of young Swiss men as assessed by 3D laser-based photonic scans and by manual anthropometric measurements"

_PeerJ, doi:10.7717/peerj.8095_

## Round 0.1 · original submission · Minor Revisions

Apologies for the lengthy delay in obtaining reviews over the summer holidays. We now have 3 constructive reviews and they broadly agree that moderate revisions are needed. Numerous helpful points are made to improve the presentation and strengthen the statistics. Please address them all, with a point-by-point Rebuttal. I will then decide if re-review is necessary or not. Thank you!

·

Basic reporting

.

Experimental design

.

Validity of the findings

.

Additional comments

Introduction: An over-the-top introduction is not the focus of the research problem.

The work is over-extensive on the basis of the objectives presented.

We should know which method of reference is: What is the anthropometry or the scanner?

As always in comparative studies, correlation coefficients are not sufficient or adequate. In this case only with the concordance study of Bland Altman sufficient, contributing the differences and its 95% CI.

I honestly think there are minimal differences between anthropometry and scanning, you don't think they have much influence on BMI.

Reviewer 2 ·

Basic reporting

Some improvements in grammar needed, particularly in the introduction.

I would suggest the addition of discussion into other 3D-based scanning techniques such as medical imaging and infra-red. Some suggested papers are listed below:

-Norton et al. (2002), 3D whole body scanning to determine mass properties of legs
-Smith & Bull (2018), Rapid calculation of bespoke body segment parameters using 3D infra-red scanning
-Cheng et al (2000), Segment inertial properties of Chinese adults determined from magnetic resonance imaging
-Durkin & Dowling (2003), Analysis of body segment parameter differences between four human populations and the estimation errors of four popular mathematical models

Experimental design

Line 99 - please add a reference for the larger research project if possible

line 102 - would it still not be possible to do a power analysis? You mention previous studies similar to yours which could provide the basis for one

line 112-115 - you could provide some justification as to the advantages or this equipment over other, smaller machines.

line 284-287 - if possible differences exist between WC measurements due to arm positions, why did you not place the subjects in the same position for both? The 3D scanner may required a specific position, so simply use this consistently.

Validity of the findings

Authors are thorough in addressing differences between manual and scanned data, suggesting possible reasons such as patient movements, breathing in during manual measurements etc ...

Additional comments

I would like to thank the authors for submitting a straight forward validation of using 3D scanners to produce height and WC measurements. The technology sector for anthropometrics is fast-growing and provides a rapid method for obtaining good data without causing discomfort to the subject.

In addition to the comments above, I would like to make some general observations:

- the use of the bioelectric impedance machine was only mentioned in the methods, but should appear in the introduction to justify its importance and necessity.
- you make mention to the fat mass results, but do not discuss at all. Again, there is mention in the abstract, but your discussion purely mentions WC and BMI.
- please add age group (not just range) of participants in Table 2.
- I think it would be interesting to observe the clinical significance between your calculated and manually measured WC and BMI values. Do the statistical differences impact on the categorisation of WC and BMI?
- Discussion does not place the work into any type of clinical 'bigger picture'. Authors make mention to the comfort of the patient, but nothing further is added in the discussion.

·

Basic reporting

no comment

Experimental design

no comment

Validity of the findings

no comment

Additional comments

The authors present novel 3d-body scanner data to validate automated body height and width circumference measurement in comparison with gold-standard manual measurement. They found very good correspondence, accompanied by systematic biases discussed appropriately.

In general, the manuscript is sound and statistical analysis straight forward. Despite relatively low sample size, which the authors also acknowledge as a limitation of the study, the data is worth publishing and results underline reliability of 3d-body scanning in health research.

I, however, have few comments and questions which the authors should address:

1) I suggest to rephrase "visual association" as e.g. used in line 232. Something like "inspection of figure xy revealed...."

2) Left panels in Figures 1-3: Please add a description to figure legend what the lines in the plots represent (x=y? linear regression?)

3) Also figure 4 needs a legend for the three line types used.

4) I suggest to blur the participant's face in Supplementary figure 1 to avoid irritations concerning privacy.

5) in line 181 it is stated that weight was obtained from BIA. Do you mean the internal scale of the BIA instrument? Please clarify.

---

## Round 0.2 · accepted · Accept

I have checked your revisions and the Rebuttal and I am satisfied that the paper has been amended appropriately. The most critical prior review's brief main points have been addressed sufficiently, and the other reviews' more minor points as well. Nicely done. I recommend having another proofread of the grammar as I noticed some minor mistakes. But as-is, scientifically the paper seems to be markedly improved.